# Omics Approaches in Pancreatic Adenocarcinoma

**DOI:** 10.3390/cancers11081052

**Published:** 2019-07-25

**Authors:** Iranzu González-Borja, Antonio Viúdez, Saioa Goñi, Enrique Santamaria, Estefania Carrasco-García, Jairo Pérez-Sanz, Irene Hernández-García, Pablo Sala-Elarre, Virginia Arrazubi, Esther Oyaga-Iriarte, Ruth Zárate, Sara Arévalo, Onintza Sayar, Ruth Vera, Joaquin Fernández-Irigoyen

**Affiliations:** 1OncobionaTras Lab, Navarrabiomed, Complejo Hospitalario de Navarra (CHN), Universidad Pública de Navarra (UPNA) Irunlarrea 3, 31008 Pamplona, Spain; 2Medical Oncology Department, Complejo Hospitalario de Navarra, Irunlarrea 3, 31008 Pamplona, Spain; 3Clinical Neuroproteomics Unit, Navarrabiomed, Complejo Hospitalario de Navarra (CHN), Universidad Pública de Navarra (UPNA), Irunlarrea 3, 31008 Pamplona, Spain; 4Proteored-ISCIII, Proteomics Unit, Navarrabiomed, Complejo Hospitalario de Navarra (CHN), Universidad Pública de Navarra (UPNA), Irunlarrea 3, 31008 Pamplona, Spain; 5Instituto de Investigación Sanitaria de Navarra (IdiSNA), Irunlarrea 3, 31008 Pamplona, Spain; 6Grupo de Oncología Celular, Instituto de Investigación Sanitaria Biodonostia, 20014 San Sebastián, Spain; 7CIBER de Fragilidad y Envejecimiento Saludable (CIBERfes), 28029 Madrid, Spain; 8Medical Oncology Department, Clínica Universidad de Navarra, 31008 Pamplona, Spain; 9Pharmamodelling S.L., 31008 Pamplona, Spain

**Keywords:** pancreatic adenocarcinoma, ctDNA, proteomic, genomic, metabolomic, lipidomic, FFPE, tissue, body fluids

## Abstract

Pancreatic ductal adenocarcinoma, which represents 80% of pancreatic cancers, is mainly diagnosed when treatment with curative intent is not possible. Consequently, the overall five-year survival rate is extremely dismal—around 5% to 7%. In addition, pancreatic cancer is expected to become the second leading cause of cancer-related death by 2030. Therefore, advances in screening, prevention and treatment are urgently needed. Fortunately, a wide range of approaches could help shed light in this area. Beyond the use of cytological or histological samples focusing in diagnosis, a plethora of new approaches are currently being used for a deeper characterization of pancreatic ductal adenocarcinoma, including genetic, epigenetic, and/or proteo-transcriptomic techniques. Accordingly, the development of new analytical technologies using body fluids (blood, bile, urine, etc.) to analyze tumor derived molecules has become a priority in pancreatic ductal adenocarcinoma due to the hard accessibility to tumor samples. These types of technologies will lead us to improve the outcome of pancreatic ductal adenocarcinoma patients.

## 1. Introduction

Pancreatic ductal adenocarcinoma (PDAC) is the seventh leading cause of cancer-related deaths worldwide. Globally, 458,918 new cases of PDAC have been reported in 2018 [1]. The expected incidence of PDAC will meaningfully increase during the present year, being the fourth leading cause of cancer deaths based on data of the American Cancer Society. Unfortunately, PDAC continues to have the lowest five-year overall survival rate—still standing at 9% [2,3]. Based on this daunting scenario, PDAC is expected to become the second leading cause of cancer deaths by 2030 [4,5], partly due to less than 20% of patients being considered as resectable at time of diagnosis, with more than half of all patients suffering distant metastases at the moment of diagnosis [6,7]. There is a clear lack of prognostic and predictive biomarkers in PDAC. CEA (carcinoembryonic antigen), a glycoprotein involved in cell adhesion circulating at very low levels in healthy adults, is increased in 30% to 60% of PDAC patients, which is significantly related to poor prognosis; however, CEA is not recommended by the current NCCN (National Comprehensive Cancer Network) guideline based on its low specificity [8]. High levels of CA125 (cancer antigen 125) are strongly associated with PDAC metastasis. It is being used in combination with CA 19-9 (carbohydrate antigen 19.9) to discriminate even early-PDAC from benign diseases [9,10,11]. Nowadays, the latter is the only biomarker recommended by the American and European guidelines for PDAC [12,13]. These guidelines recommend checking CA 19-9 levels in all patients before surgery or treatment, after resection and every 3–6 months until two years after surgery [14]. However, it should be taken into account that this carbohydrate antigen is also secreted by normal biliary epithelium and can be increased in cases of biliary obstruction, cholangitis or pancreatitis CA 19-9 has been tested for screening without remarkable results due to its reduced predictive value.

The structure of pancreatic tissue is highly complex, with exocrine (80% to 85%) and endocrine compartments (2%) that regulate the secretion of a variety of molecules (hormones, enzymes, etc.) [14]. As previously stated, the poor prognosis of PDAC is associated with its early dissemination and metastatic abilities. On one hand, this means that in a very initial stage the pancreatic tumor cells can enter into the blood stream by passive shedding, or by active means; for example, through increased motility driven by epithelial-to-mesenchymal (EMT) transition. PDACs produce and secrete a wide variety of biological molecules into the blood stream, including circulating tumoral DNA (ctDNA), exosomes, metabolites and proteins. All of them are currently being tested as potential novel biomarkers [15,16,17,18]. In addition to blood, any kind of body fluids are being considered as a potential source of diagnostic, predictive and/or prognostic biomarkers in PDAC (such as pancreatic juice, urine, pancreatic cyst fluid and bile) [19,20,21,22,23]. Here we focus on the significant findings that have been recently published on ctDNA, metabolomics/lipidomics and proteomics-based approaches, as potential sources for diagnostic, prognostic biomarkers and markers of response.

## 2. Genomics

### 2.1. Mutational Landscape and ctDNA

Since their first description in 1948, the presence of cell free DNA (cfDNA) in bloodstream is a well-established phenomenon in cancer patients and in healthy subjects [24]. Hence, circulating tumor DNA (ctDNA) can be identified in any body fluids, which can be derived from primary tumors, metastatic lesions or CTCs (Circulating Tumour Cells) [25]. In many cancer types, ctDNA is elevated in more than 85% of patients with advanced stage, although it can also be detected in early stages of cancer [17]. More importantly, several studies confirm a very strong concordance between ctDNA and tissue mutations. For example, Cohen et al. found a 100% concordance in KRAS mutation status between ctDNA and pancreatic tumoral tissue [17]. In a similar scenario, another study established that 65.8% of patients had at least one alteration in KRAS, TP53, SMAD4, or CDKN2A found in ctDNA—in high concordance with tumor tissue [26].

Currently a wide variety of methods are being used to detect and characterize ctDNA—most of them based on targeted approaches such as PCR amplification and next generation sequencing (NGS). These techniques reach a range of sensitivity of 0.01% or lower [27]. Briefly, targeted approaches rely on amplification of well-known mutations and their quantification by fluorescence or sequencing [28]. Such technologies include BEAMing (beads, emulsion, application and magnetics), which consists on a highly sensitive PCR combining emulsion PCR and flow cytometry for the identification and quantification of specific mutations [29]. dPCR (digital polymerase chain reaction) is an end-point detection technique where the absolute quantification of low expression mutations is achieved, after sample partitioning into sub-volumes [30]. Accordingly, ddPCR (droplet digital polymerase chain reaction) utilizes a similar approach where up to 20,000 amplicons can be monitored simultaneously [31]. The disadvantage of these techniques is that they rely on the prior knowledge of the specific genetic characteristics to amplify the specific sites of interest. Nevertheless, it is highly sensitive, fast, cost-effective and specific [32]. In contrast, unbiased non-targeted sequencing techniques such as whole genome (WGS) or whole exome (WES) sequencing are expensive, have lower sensitivity and require an important amount of sample [33].

Recent PDAC exome sequencing has confirmed that 95% of PDACs have KRAS mutations [34]. Several studies have used the aforementioned targeted approaches to detect different KRAS mutations (G12V, G12D, and G12R) in ctDNA as prognostic markers. For instance, using ddPCR and targeted NGS, these genetic alterations were detected in 31% to 62.6% of the PDAC cases and were correlated with shorter survival [35,36,37]. Another study using target enrichment PCR followed by NGS in a cohort of almost 200 patients with unresectable disease found a significant correlation between KRAS mutation, time-to-progression (TTP) and overall survival (OS) [38]. Another similar study detected KRAS mutation in 71% of the cases, confirming its prognostic value as predictor of progression-free survival (PFS) and OS [39].

Several groups have recently jointly analyzed DNA from exosomes (exoDNA) and ctDNA in order to ameliorate the prognosis and stratification in localized and metastatic PDAC. For instance, a recent study showed that the increase in exoDNA levels after neoadjuvant therapy was significantly correlated with earlier disease progression. In the same study, after multivariate analysis, a mutant allele fraction (MAFs) ≥5% in exoDNA was a significant predictor of PFS (HR (hazard ratio), 2.28; 95% CI (confidence interval), 1.18–4.40; *p* = 0.014) and OS (HR, 3.46; 95% CI, 1.40–8.50; *p* = 0.007). Interestingly, detection of both baseline ctDNA and exoDNA MAFs ≥5% showed to be a significant predictor of OS (HR, 7.73, 95% CI, 2.61–22.91, *p* = 0.00002) [40]. Even though KRAS mutation is the gold standard for ctDNA analyses, the advances in sequencing technologies favoring the whole PDAC genomic landscape are expanding the panel of mutations that might be detected. Accordingly, a recent study performed whole-exome sequencing in a panel of 60 hotspot genes to find candidate genetic mutations in metastatic PDAC. Then, KRAS, BRCA2, KDR, EGFR and ERBB2 were prospectively chosen for further validation by ddPCR. Interestingly, ERBB2 exon 17 mutation and KRAS G12V mutation were significantly associated with worse OS [41]. Berger et al. performed NGS and ddPCR to dynamically monitor the most frequently mutated genes in PDAC (TP53, SMAD4, CDKN2A, KRAS, APC, ATM and FBXW7) [18]. The most predominant mutations were that of KRAS and TP53. Although the combined detection levels of both were significantly decreased during treatment, a significant increase was observed during progression, which significantly correlated with PFS during therapy [18].

### 2.2. Transcriptomics

The mutational landscape of pancreatic adenocarcinoma has been extensively studied. However, transcriptome analyses have been performed in the last decade to reach a more precise molecular classification of PDAC. Already in 2008, Jones et al. analyzed the transcriptome of 24 samples of pancreatic cancer using SAGE (serial analysis of gene expression) and provided a set of differentially expressed genes comparing to micro-dissected normal pancreatic duct cells [42]. These gene sets were further validated in primary tumor-derived cell lines, opening a new avenue for the molecular classification of PDAC.

Several studies have characterized transcriptional profiles for the classification of PDAC. Collisson et al. for instance, performed microarray analysis of 27 microdissected tumors and classified PDAC into classical, quasimesenchymal (QM) and exocrine-like subtypes [43]. This was further validated by Moffitt et al. after analysis of gene expression patterns of a cohort of microarray data from 145 primary and 61 metastatic PDAC tumors, 17 cell lines, 46 pancreas and 88 adjacent normal samples [44]. They devised a classification partially overlapping that of Collison et al. [43], including classical and basal-like subtypes, and defined the importance of stroma with the identification of two subtypes: the activated-stroma (with the worst prognosis) and normal-stroma. Transcriptomic analysis might also be useful to predict sensitivity to anticancer drugs, in agreement with the studies by Duconseil et al. [45]. Cells obtained by endoscopic ultrasound guided fine-needle aspiration (EUS-FNA) and patient derived tumors were xenografted into mice. The cell lines obtained from these xenografts were subjected to different drugs and their response correlated with gene expression profiles, obtaining valuable information about patient responsiveness to drugs [45]. More recently, Bailey et al. defined four subtypes of PDAC after performing RNA-seq from 96 tumours: squamous, pancreatic progenitor, immunogenic and ADEX (aberrantly differentiated endocrine exocrine) [46].

The Cancer Genome Atlas Research Network (TCGA) also analyzed PDAC genomic and transcriptomic features [47]. 150 PDAC samples were analyzed, discriminating samples of high-purity and samples of low-purity. The strong overlap between high-purity samples with the previously reported data from Bailey [46], Collison [43] and Moffitt [44], suggested a consistent classification into basal-like/squamous and classical/progenitor groups. However, the strong association of immunogenic and ADEX or exocrine-like subtypes with the low-purity samples, suggests that these subtypes might include gene expression from non-neoplastic cells [47].

As a consequence of the increasing amount of publicly available transcriptomic data, several studies have conducted their own analyses of different RNA-seq and microarray datasets. For instance, the study from Birnbaum et al. analyzed almost 600 PDAC samples, identifying a 25-gene prognostic classifier [48]. This 25-gene classifier was associated with post-operative OS independently of classical prognostic factors and molecular subtypes. In agreement with this, other groups have identified potential prognostic biomarkers [49]. Zhao et al. included 11 datasets (including those from Collison [43], Bailey [46] and TCGA [47]), using almost 800 PDAC samples as the discovery cohort [50]. With these data the authors identified six PDAC subtypes and provided a more complete classification.

More recently, Puleo and colleagues redefined a more precise classification of PDAC after analyzing more than 300 PDAC samples. The molecular subtypes characterized are in agreement with those published until now, and include the following subtypes: pure classical, immune classical, desmoplastic, stroma activated and pure basal-like [51]. This classification allows an integrated stratification using tumoral and stromal compartments, giving special importance to the tumor microenvironment.

## 3. Proteomics

Owing to its experimental simplicity and capacity to process large cohorts of samples, label-free quantification is considered a method of choice in PDAC proteomics [52,53,54] (Table 1). The advances in mass spectrometry have allowed the simultaneous quantification of thousands of proteins. At the beginning, in-gel digestion approaches were widely used, although its low reproducibility led to the development of other tools, such as strong cation exchange (SCX) and basic pH reverse-phase (BPRP) fractionation columns with significant better sensibility and accuracy. Labeling methods by isotope-coded affinity tag (ICAT), isobaric tag for relative and absolute quantification (iTRAQ) or tandem mass tag (TMT) have been developed to increase the number of samples that can be quantified in a single MS experiment. These approaches allow the labelling of up to 3, 4 or 8 and 11 samples, respectively [55].

### 3.1. Plasma/Serum

The main issue of plasma/serum proteomics is that these samples contain a complex composition of proteins (with a high dynamic range) of many origins [77], which makes the detection of tumor-derived proteins rather challenging, as these are diluted in plasma [59]. Nevertheless, an increasing number of proteomic studies are focusing on the discovery of new plasma biomarkers for PDAC.

Recently, a multi-marker panel including apolipoprotein A-4 (APOA4), tissue inhibitor of metalloproteinase-1 (TIMP1) and CA 19-9 was shown to differentiate pancreatitis from early PDAC [56]. Another screening study of serum samples from patients with PDAC, chronic pancreatitis (CP), benign biliary disease, type 2 diabetes mellitus and healthy subjects by iTRAQ labeling [57] showed that low levels of thrombospondin-1 (THBS1) up to 24 months prior to diagnosis were associated with PDAC. Moreover, its combination with CA 19-9 showed significantly improved early diagnosis of PDAC [57]. In agreement with this, it was described that thrombospondin-2 (THBS2) discriminated among all stages of PDAC after a mass spectrometry (MS) discovery phase and confirmation by ELISA; Similarly to the previous article, THBS2 and CA 19-9 together were shown to improve the detection of patients with high risk of developing PDAC [58]. Thus, the comparative analysis of protein levels in serum samples of patients with PDAC and high-risk population could help to identify serum biomarkers for earlier diagnosis. Focusing on this line of research, a proteomic study was performed with plasma samples from PDAC patients and non-neoplastic cases (normal cases and patients with CP) [59]. After the depletion of high-abundant proteins, mass-spectrometry was performed, which led to the identification of 1340 plasma proteins. Again, nine mediator proteins were validated by ELISA; metalloproteinase inhibitor 1 (TIMP1), intercellular adhesion molecule 1 (ICAM1), thrombospondin-1 (THBS1), C-C motif chemokine 5 (CCL5 or RANTES), lipopolysaccharide-binding protein (LBP) and platelet basic protein (PPBP). Other three candidates were identified for CP, such as zinc-alpha2-glycoprotein 1 (AZGP1), apolipoprotein A-II (APOA2) and lactotransferrin (LTF). Importantly, TIMP1 levels distinguished PDAC from controls [78]. Most of these proteins were involved in interleukin 4, 10 and 13 pathways. Instead of depleting proteins using a column, a combination of two strategies, such as antibody-based proteomics and liquid chromatography-tandem mass spectrometry (LC-MS/MS), can be used [60]. For instance, a study using this methodology found that insulin-like growth factor-binding protein-2 (IGFBP2) and -3 (IGFBP3) are increased and decreased, respectively, in plasma of early-stage invasive ductal PDAC) [60]. These factors together with CA 19-9 could enable a more reliable diagnosis of early-stage PDAC.

The use of high-definition mass spectrometry (HDMSE) in the characterization of the proteome of three groups of patients (resectable PDAC, controls and patients with benign pancreatic diseases) identified 715 proteins [61]. From these, 40 of them were significantly overexpressed in PDAC and were related to cell cycle, apoptosis, cell adhesion, cell chemotaxis and immune response process [61]. In a similar study, CXC chemokine ligand 7 (CXCL7, or NAP2) was significantly decreased in plasma samples of PDAC patients [62].

Pathways related with extracellular matrix remodeling were found to be enriched in all datasets of differentially expressed proteins in PDAC. In summary, human plasma samples can be successfully processed to search for potential biomarkers after removing high-abundance proteins. Moreover, most of these recent studies suggest that CA 19-9 should be used in combination with multiprotein panels to achieve a better diagnostic accuracy.

### 3.2. Pancreatic Juice

Pancreatic juice has been studied as a source of proteins in PDAC [20]. The proteome of pancreatic juice has been recently characterized in patients with PDAC [63], as well as in premalignant lesions and benign pancreatic diseases [64,65].

Using isotope-code affinity tag (ICAT) technology and MS/MS, differentially expressed proteins were detected in PDAC samples compared to control cases, most of them involved in insulin regulation and catalytic metabolism [63]. Others have used in-gel electrophoresis (DIGE) and MS/MS, identifying differently expressed proteins in PDAC patients. This protein subset was further confirmed by western blot, highlighting matrix metalloproteinase-9 (MMP-9), which was significantly increased in serum of PDAC patients [64].

iTRAQ quantitative proteomics was used to characterize the proteome of pancreatic juice samples from patients with primary pancreatic intraepithelial lesion (PanIN) 3. They found that anterior gradient-2 (AGR2) was overexpressed in the juice of PanIN3 cases compared to non-diseased controls. Moreover, this result was validated in an independent cohort of patients with PDAC, pre-malignant lesions (including PanIN3, PanIN2 and Intraductal Papillary Mucinous Neoplasia-IPMNs) and benign pancreatic diseases, showing that AGR2 levels were significantly elevated in both cases (PDAC and any grade of dysplasia) compared to control samples [65]. This finding also correlated in tissue samples from a different study, in which AGR2 was found to be overexpressed in tissues of PanIN and PDAC patients [79]. These studies suggest that AGR2 could have a potential value as pancreatic juice biomarker for PDAC early detection.

### 3.3. Pancreatic Cyst Fluid

Most PDAC cases derive from three types of lesions: PanIN, IPMN and mucinous cystic neoplasm [80]. In the latter, pancreatic cyst fluid can be another source for early diagnostic biomarkers [20]. Applying 2-D electrophoresis (2-DE) and LC MS/MS to cyst fluid samples from 20 patients, a panel of potential protein biomarkers was found, including two homologs of amylase, solubilized molecules of four mucins, four solubilized CEA-related cell adhesion molecules and four S100 homologs [22]. Despite being a widely used technique, 2-DE has weaknesses related to poor detection of low abundant proteins, separation, resolution of hydrophobic proteins, difficulties with proteins >150 kDa and issues with proteins with extreme isoelectric points (pIs) [81].

### 3.4. Urine

Contrary to what might be expected, recent studies have described certain proteins in urine that could have a role as potential biomarkers for PDAC [82]. An analysis of 18 urine samples derived from PDAC, CP and healthy individuals by gel electrophoresis followed by LC and MS/MS (GeLC/MS/MS) identified around 1500 proteins. Although sex-dependent differences were observed, three proteins were deregulated in both males and females: lymphatic vessel endothelial hyaluronan receptor 1 (LYVE1), regenerating gene (REG1A) and trefoil factor (TFF1). The high expression of these proteins in PDAC was subsequently confirmed by ELISA, indicating that this group of proteins distinguished early PDAC from healthy samples with high accuracy [66].

Other studies pointed out D-dimer as a predictor of tumor resectability in PDAC. D-dimer is the final degradation product of crosslinked fibrin and it was measured in several body fluids such as peripheral blood, bile, urine and portal blood in resectable and unresectable PDAC patients [67]. D-dimer was present at different concentrations, being lower in unresectable patients. Weeks et al. detected 101 differentially expressed proteins between PDAC and PC patients by MALDI-TOF MS, highlighting CD59 glycoprotein (CD59), annexin A2 (ANXA2), 21 kDa gelsolin (GSN) fragment, protein S100-A9 (S100A9) and tumor necrosis factor alpha-induced protein 3 (TNFAIP3), which have been found to be overexpressed in PDAC tissues [21]. Similarly, another study differentiated PDAC and CP samples by capillary electrophoresis-MS analysis by choosing the most stable urinary peptides, with a significant accuracy improvement compared to CA 19-9 quantification [68]. All these data suggest that urine samples have a significant potential for discovery of PDAC-related biomarkers.

### 3.5. Bile

Bile duct stenosis can occur in benign or malignant pancreatic conditions such as CP and PDAC, respectively. Farina et al. performed a proteomic analysis of bile samples from patients with bile duct stenosis caused by PDAC. After SDS-PAGE and LC-MS/MS 127 proteins were identified, some of them previously described as potential biomarkers for PDAC [69]. Carcinoembryonic antigen-related cell-adhesion molecule 6 (CEACAM6) was increased in bile in five out of six cases with PDAC, which correlated with a previously reported function of CEACAM6 as a potential prognosis biomarker in PDAC [83].

### 3.6. Pancreatic Tissue

PDAC is mainly characterized by its vascular deficiency and an abundant desmoplastic stroma, which usually represents 90% of tumor volume. The stroma is comprised of cancer-associated-fibroblasts, inflammatory cells, embedded in a rich and dense extracellular matrix [84,85,86]. Proteomic studies have been performed not only in fresh tissue, but also in preserved tissue such as formalin-fixed paraffin-embedded (FFPE) tissue. Accordingly, when fresh tissue is not available, FFPE tissue blocks are used to perform molecular analysis by mass spectrometry [87].

Using a label-free approach through LC-MS/MS, 99 differentially expressed proteins were found between tumor tissue and adjacent non-tumor tissue, where prolargin (PRELP) seemed to have a potential prognostic role in cases where surgery was performed [53]. Previously, another study using iTRAQ labeling found that dihydropyrimidinase-like 3 (DPYSL3) was significantly differentially expressed in PDAC tissue compared to normal tissue [70].

Similarly, another study performed by quantitative glycoproteomics found more than 600 N-glycopeptides derived from 374 N-glycoproteins [71]. Interestingly N-glycosilation levels were increased in pancreatic cancer samples,] from proteins such as mucin-5AC (MUC5AC), carcinoembryonic antigen-related cell adhesion molecule 5 (CEACAM5), insulin-like growth factor binding protein 3 (IGFBP3) and galectin-3 binding protein (LGALS3BP) [71]. Another proteomic study using 10 cases of PDAC (matching normal and tumor tissues) had previously confirmed the high correlation of galectin-1 with grade, T and N stage [72].

Other authors [73] have conducted quantitative proteomic studies on micro-dissected premalignant lesions (PanIN tissue). After 2-DE analysis, differentially expressed proteins were identified using nanoLC-ESI-MS/MS in different grade PanIN lesions (PanIN 1A/B, PanIN 2, PanIn3) from nine patients. Eighty-six proteins were significantly regulated during PanIN progression, which were further validated by immunohistochemistry. These results highlighted major vault protein (MVP), anterior gradient 2 (AGR2), 14-3-3 sigma, annexin A4 and S100A10 as differentially expressed in PanIN lesions [73].

Joao A et al. [74] performed two studies using FFPE tissue. Firstly, they compared normal, CP and PDAC tissue using LC MS/MS. The authors [74] found some exclusive proteins for normal, CP or PDAC specimens. Accordingly, annexin 4A was only found in PDAC and was associated with epithelial cell differentiation and negative regulation of apoptotic processes. Similar results were found with fibronectin, involved in angiogenesis and cell adhesion pathways [74]. Secondly, a similar approach was performed in autoimmune pancreatitis, CP and PDAC samples using LC MS/MS [75]. Proteins found exclusively in PDAC included epiplakin, mucin2 (MUC2) and protein disulfide-isomerase A3. Epiplakin is expressed by centroacinar and pancreatic duct cells in precursor lesions, which had been previously described in in vivo models, [88]. MUC2 is a well-known glycoprotein with an important role in ductal architecture, also expressed in precursor lesions [89]. Protein disulfide-isomerase A3 has been described in Alzheimer´s disease and PDAC pathogenesis, and regulates the cleavage of disulfide bonds between cysteine nucleotides [90].

Following this, Naidoo et al. performed the first study comparing the proteome of PDAC primary tumors and matched lymph node metastases, with samples obtained by laser capture microdissection from FFPE [76]. The authors identified 1504 proteins in PDAC primary tumors and compared then with previous proteomic dataset obtained from pancreatic tissue, pancreatic juice, serum and urine [76]. The authors showed that 30% of these proteins overlapped, expanding the database of specific PDAC proteins. Most of the significant pathways associated to these proteins were related to immune system, metabolism, signal transduction and metabolism of proteins [76]. Thirteen percent of the proteins had significant differential expression, including S100P (with important role in endothelial cell migration and neutrophil degranulation) and 14-3-3 sigma (involved in DNA damage response). These latter proteins were validated in an independent cohort.

A proteome-scale interaction network with the differentially regulated proteins identified by tissue-proteomics was performed using STRING software (Figure 1) [91]. To minimize false positives and false negatives, only interactions tagged as “highest confidence” (>0.9) in the STRING database were considered. The topological analysis of this network uncovered multiple subnetworks that are functionally related. We found that there were more up-regulated than down-regulated proteins, finding TPM4 (P67936) and ALDH1A1 (P00352) shared in both cases. The networks of up-regulated proteins consisted on various subnetworks (Figure 1A,B). One of them (right) has two common functionalities; neutrophil degranulation and regulated exocytosis. Neutrophils participate in cancer by releasing their granular content and exerting anti-inflammatory responses [92]. Accordingly, neutrophil-to-lymphocyte ratio is a classical prognostic factor of OS. Moreover, neutrophil granular proteins have been associated with tumor progression, including some cytokines that can be tracked in blood samples [93]. The other protein network subset (left) is mainly involved in ribosomal and mRNA metabolism. Ribonucleoproteins participates in gene expression regulation, transcription, RNA alternative splicing, translation and posttranslational modification. Some ribonucleoproteins have been associated with tumor progression [94]. For example, high ribonucleoprotein A1 (HNRNPA1) expression correlates with poor OS in hepatocellular carcinoma [95].

## 4. Metabolomics and Lipidomics

Metabolomics is based on the characterization and quantification of low molecular weight biological compounds (lipids, amino acids, alcohols, carbohydrates, etc.) that can be found in fluids or tissues. The most widely used technology for the unbiased characterization of metabolites is liquid-chromatography followed by mass spectrometry (LC-MS). Because of the deregulated metabolism of cancer cells, studies characterizing cancer metabolome as diagnostic and/or potential predictive biomarker are increasing. Hence, LC-MS was performed in blood samples from PDAC and CP patients [96]. From 4578 metabolites, glycocholic acid, hexanoylcarnitine and N-palmitoyl glutamic acid were significantly upregulated in PDAC and discriminated PDAC from CP patients [96]. Another study showed a biomarker signature of CA 19-9 and a serum/plasma panel of nine metabolites (proline, sphingomyelin, phosphatidylcholine, isocitrate, sphingamine1-phospate, histidine, pyruvate, ceramide and sphingomyelin), which discriminated PDAC from PC patients at diagnosis [97].

Following the trend of early detection in PDAC, a study using ^1^H NMR and 2D NMR spectroscopy in 2011 found differences in the metabolomic profile between PDAC patients and subjects with benign disease [98]. Accordingly, concentrations of glutamate, acetone and 3-hydroxybutyrate were higher in PDAC than in benign disease [98]. More recently, a six-metabolite panel could differentiate early stage PDAC from benign pancreatic disease [99]. Other studies have focused on the potential predictive value of metabolic biomarkers in relation with response/resistance to a treatment. Phua et al. found 19 potential markers (lactic acid, proline and pyroglutamate among others), that could be predictive of gemcitabine chemoresistance in PDAC patients [100].

Similar to metabolomic platforms, lipidomics rely on the identification and quantification of lipid metabolites, for which mass spectrometry is commonly used [101] together with traditional procedures for sample preparation [102,103], or improved methods based on a mixture of methanol /chloroform/ methyl tert-butyl ether (MTBE) or ultrasound applications [101]. Lipids can be classified into eight categories: fatty acids or acyls (FAs), glycerophospholipids (GPLs), glycerolipids (GLs), prenol lipids (PRLs), saccharolipids (SCLs), sphingolipids (SLs) and sterol lipids (STLs) [104]. As early as 2002, it was reported that plasma concentrations of n-3 fatty acids (FA), phospholipids and cholesteryl esters were lower in PDAC than in healthy subjects [105], similarly to what has been confirmed recently [106]. Jiang et al. applied LC-MS/MS in PDAC tissue plasma samples [107], finding higher levels of ceramide species in tumor samples with lymph node metastases compared to patients without lymph node dissemination [107]. In addition, some cytokines, such as interferon inducible protein 10 (IP10) and epidermal growth factor (EGF), were directly correlated with several sphingolipid (SL) variants.

## 5. Conclusions

The technological advances of omics platforms and bioinformatic analyses have favoured a greater knowledge of the biological behaviour and dynamics of a variety of neoplastic entities including PDAC [10,108,109,110], favouring earlier diagnosis in asymptomatic patients [111,112]. In a time of special transcendence in oncology, PDAC remains to be a problem of significant magnitude. First; because of its progressive increase in incidence, and second; for its aggressiveness and scarcity of curative options. In fact, in spite of these technological advances, PDAC prognosis today has hardly changed compared to decades ago, unlike other oncological malignancies. The prognostic improvement over different neoplastic entities has been caused by the spectacular development of targeted therapies and, above all, immunotherapy. Unfortunately, PDAC is one of the malignancies for which these treatments are not proving significant benefit. Only in cases where DNA repair machinery is altered, due to the loss of function of MMR proteins (MLH1, MSH2, MSH6 and PMS2) or the presence of mutations in BRCA1/BRCA2 [113], immunotherapy or PARP inhibitors such as olaparib [114,115] have shown significant efficacies in limited cases [116,117,118,119,120].

In PDAC, the failure of the main types of immunotherapies might be explained on one hand by its low immunogenicity and low mutational load [121,122]. On the other hand, as previously explained, by the presence of a peritumoral “stromal-desmosplastic rock”, which seems to favour immunosuppression through the infiltration by suppressive myeloid cells [123]. The development of new treatment strategies based on the activation of cytotoxic T cells (through radiation, chemotherapy, vaccines, CD40 agonists, JAK or PI3K inhibitors, etc.) or in the weakening of this exacerbated stromal reaction (by degradation of hyaluronan using pegvorhyaluronidase alfa-PEGPH20) [113], is expected to have benefits. More importantly, the integration of multi-omic platforms data is believed to offer a wide range of useful tools for the characterization of PDAC tumor biology and research in diagnosis/prognostic/predictive biomarkers [124]. This includes not only through the analysis of proteins, but also through the circulating tumor nucleic acids or extracellular vesicles such as exosomes. Hence, exosomes are released into the blood [125] and may have a varying composition in proteins, mRNA, DNA, etc. The detailed study of these molecules through multi-omic approaches might offer relevant information to cancer progression and angiogenesis (through vascular endothelial growth factor (VEGFR) or fibroblast growth factor (FGF)) [15,125], EMT (vimentin, caveolin-1 or IL-6) [126], metastasis (macrophage migration inhibitory factor (MIF), miR-17-5p) [127,128], and/or drug resistance [129]. Proteomic approaches have deciphered the exosomal protein composition, using iTRAQ labeling for example. Indeed, these studies have detected up to 800 proteins, some of which are implicated in metastasis and drug resistance [130]. Taking this into consideration, multi-omic aproaches and the integration of this “big data”, may offer relevant information on the biology of PDAC, leading us to the discovery of different types of biomarkers [131]. Cancer integration via multikernel learning (CIMLR) or proteogenomics are new methods that integrate multi-omic data, and have been used in several neoplasias—but not in PDAC [132,133,134].

In summary, we are hopeful that the development of these omics platforms will help to improve the current outcome of a disease as ominous PDAC.

## Figures and Tables

**Figure 1 cancers-11-01052-f001:**
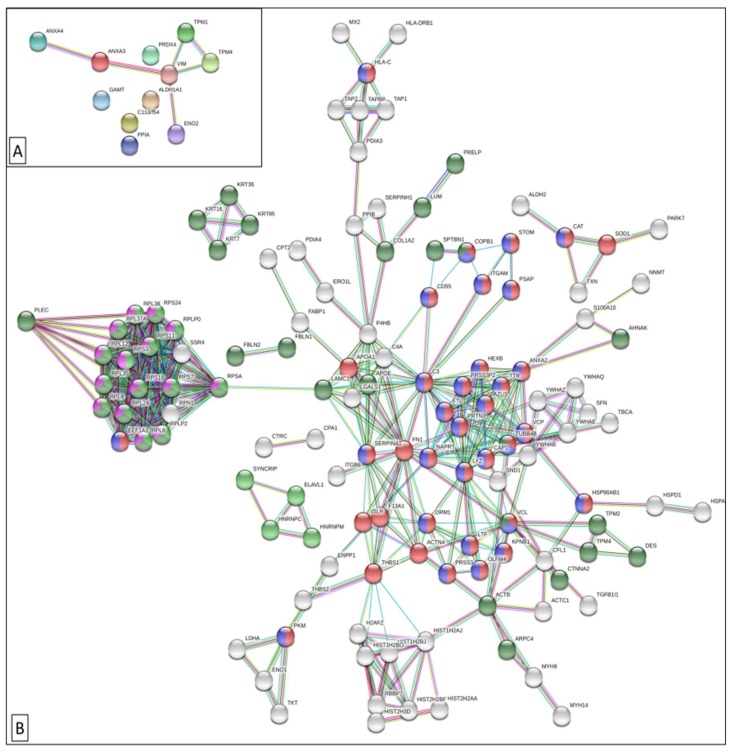
Protein functional interactome network for differentially expressed proteins identified in Pancreatic ductal adenocarcinoma (PDAC) tissue; down-regulated proteins (**A**) and up-regulated proteins (**B**). Using STRING software, proteins are represented as nodes, and interactions with continuous lines representing functional association. The closets subsets of proteins share functionalities. In the figure, some functional pathways have been highlighted with color codes representing proteins that belong to the same pathway. Regulated exocytosis (red), mRNA metabolic process (green), ribosomal (purple), neutrophil degranulation (blue) and structural molecule activity (dark green).

**Table 1 cancers-11-01052-t001:** Summary of proteomic studies in PDAC listed by the type of sample used in different proteomic approaches.

Technology	Ref.	Sample Type	Summary	Potential as	Pros/Cons
SID-MRM-MS	[56]	Serum	A panel of APOA4, TIMP1 and CA 19-9 could differentiate pancreatitis from early PDAC.	Diagnostic	Detection of low abundant peptides
iTRAQ and MRM	[57]	Serum	TSP-1 decreases before the diagnosis of PDAC and low levels were associated with poor prognosis	Prognosis	Analyze different samples and compare them in the same assay
HPLC-SCX-MS and ELISA	[58]	Plasma	TSP-2 could discriminate PDAC patients from healthy controls	Diagnostic	Peptides elute according to their affinity for the columns giving after a higher detection of low abundant peptides
MARS-human 7 HPLC column, SCX and MS	[59]	Plasma	TIMP1 and ICAM had a better performance (AUC 0.92) than CA 19-9	Diagnostic	The most abundant proteins in circulation are depleted from samples giving after a higher detection of low abundant peptides
LC-MS/MS	[60]	Plasma	IGFBP2 and IGFBP3 could differentiate PDAC and chronic pancreatitis	Diagnostic	High sensitivity and specificity in plasma analysis
HDMSE	[61]	Plasma	40 proteins were found overexpressed in PDAC patients	Diagnostic	Enables deeper proteome coverage and more confident peptide identifications
LC-MS	[62]	Plasma	CXCL7 was significantly decreased in PDAC patients	Diagnostic	High sensitivity and specificity in plasma analysis
ICAT MS/MS	[63]	Pancreatic juice	IGFBP2 was overexpressed in PDAC	Diagnostic	Analyze different samples and compare them in the same assay
DIGE MS/MS	[64]	Pancreatic juice	MMP-9 was significantly higher in juice and serum of PDAC patients	Diagnostic	Low reproducibility
iTRAQ MS/MS	[65]	Pancreatic juice	AGR2 was overexpressed in PanIN and PDAC samples compared to control.	Diagnostic	Analyze different samples and compare them in the same assay
GeLC/MS/MS	[66]	Urine	LYVE1, REG1A and TFF1 were significantly higher expressed in PDAC.	Diagnostic	High sensitivity
VIDAS D-Dimer Exclusion II, bioMérieux	[67]	Urine	D-dimer was lower in unresectable cases (urine) but higher in other study (blood).	Disparity	Specific product for this purpose
DIGE MALDI-ToF MS	[21]	Urine	CD59, ANAx2, GSN, S100A9 and TNFAIP3 were overexpressed in PDAC urine and tumor sample.	Diagnostic	low reproducibility
CE-MS	[68]	Urine	PDAC and chronic pancreatitis were identifiable using 47 candidate biomarkers	Diagnostic	High sensitivity
SDS-PAGE LC-MS/MS	[69]	Bile	CEACAM6 was increased in PDAC cases (bile and tissue) that correlated with shorter OS.	Prognosis	low reproducibility
LC-MS/MS	[53]	Tissue	99 proteins were differentially expressed. PKELP was verified	Diagnosis	High sensitivity
LC-MS/MS	[70]	Tissue	DPYSL3 as the best diagnostic marker	Diagnosis	High sensitivity
LC-MS/MS	[71]	Tissue	MUC5AC, CEACAM5, IGFBP3 and LGALS3BP have aberrant N-glycosylation levels associated with pancreatic cancer.	Diagnosis	High sensitivity
SDS-PAGE MALDI-TOF MS	[72]	Tissue	Galectin-1 correlated with histology, T stage and N stage	Diagnosis/prognosis	Low reproducibility
nanoLC-ESI-MS/MS	[73]	Tissue	MVP, AGR2, 14-3-3 sigma, annexin A4 and S100A10 were differentially expressed in PanIN lesions.	Predictive /diagnosis	High sensitivity
LC MS/MS	[74]	FFPE	Annexin 4A and fibronectin were only detected in PDAC cases.	Diagnosis	High sensitivity
LC MS/MS	[75]	FFPE	Epiplakin, MUC2, protein disulfide-isomeraseA3 were exclusively detected in PDAC.	Diagnosis	High sensitivity
LC-ESI-MS/MS	[76]	FFPE	13% of proteins were differentially expressed. S100P and 14-3-4 sigma were validated.	Diagnosis	High sensitivity

Capillary electrophoresis–mass spectrometry (CE-MS); difference gel electrophoresis (DIGE); electrospray ionization mass spectrometry (ESI-MS); enzyme-linked immuno sorbent assay (ELISA); high performance liquid chromatography (HPLC); isobaric tags for relative and absolute quantitation (iTRAQ); isotope-coded affinity tag (ICAT); liquid chromatography–mass spectrometry (LC-MS); tandem mass spectrometry high definition MSE (HDMSE); matrix-assisted laser desorption/ionization time-of-flight. (MALDI TOF); one-dimensional sodium dodecyl sulfate-polyacrylamide gel electrophoresis followed by liquid chromatography-tandem mass spectrometry (GeLC-MS/MS); stable isotope dilution coupled with multiple reactions monitoring mass spectrometry (SID-MRM-MS); strong cation-exchange (SCX); Formalin-Fixed Paraffin-Embedded (FFPE); Pancreatic ductal adenocarcinoma (PDAC).

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
