# Peer review of "Omics Approaches in Pancreatic Adenocarcinoma"

_cancers, 2019, doi:10.3390/cancers11081052_

Round 1

Reviewer 1 Report

The authors reviewed very recent discoveries of PDAC biomarkers with omics-related techniques in this manuscript. It includes three parts: genomics, proteomics and metabolomics/lipidomics. Among them, proteomics development and applications on PDAC biomarkers discovery are highlighted, and described very comprehensively in different types of biological samples. Overall, it is a well-written review paper, and could be suitable be published after minor revisions.

 1. For proteomics technique development, label-free quantitative proteomics cannot be avoided since it is able to analyze theoretically unlimited biological samples, which is necessary and critical for large-cohort analysis in pharmaceutical/biomedical research. Thus, several examples/discoveries of PDAC biomarkers using label-free quantitative proteomic method are suggested to be included in this paper.

2. As each omics technique mentioned in this paper is not totally independent, some new omics topics combining multiple omics techniques are being emphasized these years (e.g. genoprotoemics) and could be updated in this review.

Author Response

First of all we would like to thank the editor and the reviewers for reading the manuscript, and for considering that it can be a hypothetical interesting contribution. We then respond point by point to suggestions and recommendations for improving the text.

REVIEWER 1

The authors reviewed very recent discoveries of PDAC biomarkers with omics-related techniques in this manuscript. It includes three parts: genomics, proteomics and metabolomics/lipidomics. Among them, proteomics development and applications on PDAC biomarkers discovery are highlighted, and described very comprehensively in different types of biological samples. Overall, it is a well-written review paper, and could be suitable be published after minor revisions.

 1. For proteomics technique development, label-free quantitative proteomics cannot be avoided since it is able to analyze theoretically unlimited biological samples, which is necessary and critical for large-cohort analysis in pharmaceutical/biomedical research. Thus, several examples/discoveries of PDAC biomarkers using label-free quantitative proteomic method are suggested to be included in this paper.

Based on Reviewer 1 recommendations, we have included a paragraph at the beginning of the Proteomics section focused on label-free quantitative proteomics.

2. As each omics technique mentioned in this paper is not totally independent, some new omics topics combining multiple omics techniques are being emphasized these years (e.g. genoprotoemics) and could be updated in this review.

We agree with Reviewer 1 that in other neplasias the use of several omic platforms is already a reality, although their implementations in PDAC entail more difficulty, so we have pointed out these caveats in last paragraph of the conclusions.

Reviewer 2 Report

The authors present a comprehensive review on the utility of omics approaches for discovery of biomarkers in pancreatic cancer. This is especially important in pancreatic cancer due to our inability to detect and daignose cancers at early stages where resection is viable. This has led to a dismal 5-year survival rate of less than 10% as most patients are diagnosed only after the cancer has metastasized. The authors address the importance of discovering novel and reliable biomarkers using a plethora of omics platforms. They also discuss these markers in the context of detection, diagnosis as well as longitudinal assessment of treatments. I would, however, urge the authors to address the following points prior to publication:

1. The biggest concern I have is overlap with recently published reviews Daamen et al, 2018 [PMID: 29366815]  that discusses serum markers in cancer detection in detail.

2. Section 3.2: Although the authors briefly mention the markers discovered in pancreatic juices, they do not address the difficulty in obtaining these samples. The authors should address the methods of obtaining such samples and there viability in diagnoses.

3. Line 301, STRING analysis. Was this performed by pooling all differential proteins (both up- and downregulated)? Will the inferences change, if we consider upregulated and downregulated proteins separately?

4. Line 336, it should be made clear how the metabolic profile was obtained: via serum or tumor samples, etc?

5. Line 84, 166, 305: Typos in punctuation

Author Response

First of all we would like to thank the editor and the reviewers for reading the manuscript, and for considering that it can be a hypothetical interesting contribution. We then respond point by point to suggestions and recommendations for improving the text.

REVIEWER 2

The authors present a comprehensive review on the utility of omics approaches for discovery of biomarkers in pancreatic cancer. This is especially important in pancreatic cancer due to our inability to detect and daignose cancers at early stages where resection is viable. This has led to a dismal 5-year survival rate of less than 10% as most patients are diagnosed only after the cancer has metastasized. The authors address the importance of discovering novel and reliable biomarkers using a plethora of omics platforms. They also discuss these markers in the context of detection, diagnosis as well as longitudinal assessment of treatments. I would, however, urge the authors to address the following points prior to publication:

1. The biggest concern I have is overlap with recently published reviews Daamen et al, 2018 [PMID: 29366815]  that discusses serum markers in cancer detection in detail.

Obviously, this reference has been included in the new version of the manuscript due to significance, in spite of this systematic review is focused in role of serum tumor markers (mainly CA 19.9, CEA, imaging tests) for surveillance after surgical resection of pancreatic cancer.

2. Section 3.2: Although the authors briefly mention the markers discovered in pancreatic juices, they do not address the difficulty in obtaining these samples. The authors should address the methods of obtaining such samples and there viability in diagnoses.

Based on Reviewer 2 suggestions we halve included some more detail explanations about pancreatic juices sampling in first part of this section.

3. Line 301, STRING analysis. Was this performed by pooling all differential proteins (both up- and downregulated)? Will the inferences change, if we consider upregulated and downregulated proteins separately?

Based on your considerations, we have tried to explain it in more detail at the end of section 3.

4. Line 336, it should be made clear how the metabolic profile was obtained: via serum or tumor samples, etc?

We have tried to resolve the misunderstanding by explaining that they were serum and plasma samples

5. Line 84, 166, 305: Typos in punctuation

These typos has been solved

Reviewer 3 Report

In this review manuscript, entitled” Omics Approaches in Pancreatic Adenocarcinoma”. The authors summarized some approaches that focused on pancreatic ductal adenocarcinoma genetic, epigenetic, and/or proteo, transcriptomics materials and listed the new technologies using body fluids (blood, bile, urine, etc) where tumor derived molecules are analyzed (DNA, proteins, lipids) becomes more important detection sample types for pancreatic ductal adenocarcinoma diagnosis or prognosis. The review article is interesting and within the scope of the journal. Some concerns are: 

 1.      The authors should add Transcriptomics such as miRNA biomarkers in this review.

2.      The authors should compare the advantage and limitations in different samples collection or technologies they listed in this review.

3.       The authors should emphasize which technologies are suitable for early diagnosis, and which ones are good for prognosis?

4.      The authors may cite more references in this review.

Author Response

First of all we would like to thank the editor and the reviewers for reading the manuscript, and for considering that it can be a hypothetical interesting contribution. We then respond point by point to suggestions and recommendations for improving the text.

REVIEWER 3

In this review manuscript, entitled” Omics Approaches in Pancreatic Adenocarcinoma”. The authors summarized some approaches that focused on pancreatic ductal adenocarcinoma genetic, epigenetic, and/or proteo, transcriptomics materials and listed the new technologies using body fluids (blood, bile, urine, etc) where tumor derived molecules are analyzed (DNA, proteins, lipids) becomes more important detection sample types for pancreatic ductal adenocarcinoma diagnosis or prognosis. The review article is interesting and within the scope of the journal. Some concerns are:

 1.      The authors should add Transcriptomics such as miRNA biomarkers in this review.

During the writing of this review, we have specifically focused in ctDNA, metabolomics/lipidomics and proteomics-based approaches. Based on Reviewer 3 (and others) considerations we have included a Transcriptomic Section in this new manuscript version.

Taking into account the limitations of text length (stipulated by the editor), we believe that a section based on microRNA etc could be better adapted within other review focused in Epigenetic in PDAC.

2.      The authors should compare the advantage and limitations in different samples collection or technologies they listed in this review.

According to this consideration we included it in Table 1.

3.       The authors should emphasize which technologies are suitable for early diagnosis, and which ones are good for prognosis?

Similarly, we include a new column in Table 1

4.      The authors may cite more references in this review.

Thirteen new references have been included in this new manuscript version.

Reviewer 4 Report

In the submitted Review manuscript, Gonzalez-Borja and colleagues describe some omics approaches that will hopefully yield markers for better diagnosis of PDAC. There are some fundamental problems with the manuscript that lead me to advise against its publication.

Comments:

The review lacks balance and focus. The title suggests that the reader will be informed on the different –omics approaches that have been applied in PDAC, but instead there are large sections dedicated to liquid biopsy proteomics and ctDNA. These have their place in research but in this manuscript they feature too prominently. What about genomics, epigenomics, and transcriptomics? This is not something that cannot be easily fixed. 

Some of the sections go into excessive detail, this harms legibility of the paper as a whole.

There are numerous reviews with a similar scope.

What data are in Figure 1?  Is this original data or a meta analysis of existing proteomics data?

Figure 1 node legends are illegible.

Some English editing could be done.

Author Response

First of all we would like to thank the editor and the reviewers for reading the manuscript, and for considering that it can be a hypothetical interesting contribution. We then respond point by point to suggestions and recommendations for improving the text.

REVIEWER 4

In the submitted Review manuscript, Gonzalez-Borja and colleagues describe some omics approaches that will hopefully yield markers for better diagnosis of PDAC. There are some fundamental problems with the manuscript that lead me to advise against its publication.

 Comments:

The review lacks balance and focus. The title suggests that the reader will be informed on the different –omics approaches that have been applied in PDAC, but instead there are large sections dedicated to liquid biopsy proteomics and ctDNA. These have their place in research but in this manuscript they feature too prominently. What about genomics, epigenomics, and transcriptomics? This is not something that cannot be easily fixed.

Based on Reviewer 4 comment, we changed Section 2 that now includes Genomic subsection (mainly focused in ctDNA) and Transcriptomic Section. Mainly limited by text length, epigenomic section probably cannot be added to this work.

Some of the sections go into excessive detail, this harms legibility of the paper as a whole.

There are numerous reviews with a similar scope.

Although we agree that there are some reviews that explain the role of some omic platforms each on its own, it is difficult to find one that that compiles updated information from those different platforms, especially in pancreatic adenocarcinoma.

What data are in Figure 1?  Is this original data or a meta analysis of existing proteomics data?

The Figure 1 is a compilation of differentially expressed proteins previously detected in PDAC proteomic studies.

Figure 1 node legends are illegible.

We tried to enlarge this figure in order to legends can be more legible.

Some English editing could be done.

English edition has been done for this new version.

Round 2

Reviewer 3 Report

The authors have provided a detailed and thorough response to the comments given by reviewers. One more point is that the authors should ensure to refer  the table 1 in the main text.

Reviewer 4 Report

The authors addressed some of my concerns but the fundamental concern of novelty still stands.